# Cyclic AMP but Not Calmodulin as a Potential Wasoconstrictor in Simulated Reperfusion

**DOI:** 10.3390/ijms241210355

**Published:** 2023-06-19

**Authors:** Jakub Ohla, Michał Wiciński, Maciej Słupski, Jan Zabrzyński, Bartosz Malinowski

**Affiliations:** 1Department of Orthopaedics and Traumatology, Faculty of Medicine, Collegium Medicum in Bydgoszcz, Nicolaus Copernicus University in Toruń, 85-090 Bydgoszcz, Poland; jan.zabrzynski@cm.umk.pl; 2Department of Pharmacology and Therapeutics, Faculty of Medicine, Collegium Medicum in Bydgoszcz, Nicolaus Copernicus University in Toruń, 85-090 Bydgoszcz, Poland; michal.wicinski@cm.umk.pl (M.W.); bartosz.malinowski@cm.umk.pl (B.M.); 3Department of Hepatobiliary and General Surgery, Faculty of Medicine, Collegium Medicum in Bydgoszcz, Nicolaus Copernicus University in Toruń, 85-090 Bydgoszcz, Poland; maciej.slupski@cm.umk.pl

**Keywords:** ischemia reperfusion injury, VSMCs, cAMP, calmodulin

## Abstract

The phenomena of ischemia and reperfusion are associated with the pathological background of cardiovascular diseases. Ischemia is initiated by ischemia reperfusion injury (IRI), which involves disruption of intracellular signaling pathways and causes cell death. The aim of this study was to assess the reactivity of vascular smooth muscle cells in the conditions of induced ischemia and reperfusion, and to determine the mechanisms leading to contractility disorders. This study was conducted using classical pharmacometric methods on an isolated model of the rat caudal artery. The experiment consisted of the analysis of the final and initial perfusate pressure measurements after induction of arterial contraction with phenylephrine in the presence of forskolin and A7 hydrochloride, two ligands modifying the contractility of vascular smooth muscle cells (VSMC). The pharmacometric analysis showed that in simulated reperfusion, cyclic nucleotides have a vasoconstrictive effect, and calmodulin has a vasodilating effect. The responsiveness of vascular smooth muscle cells to the vasopressor effects of α1-adrenomimetics during reperfusion may change uncontrollably, and the effects of secondary messengers may be counter physiological. Further studies are needed to evaluate the function of other second messengers on VSMCs in the process of ischemia and reperfusion.

## 1. Introduction

Ischemia and reperfusion processes are the pathological basis of cardiovascular and central nervous system disorders. Ischemia results in disorders of cellular homeostasis resulting from the inhibition of adenosine-5′-triphosphate (ATP) production [1], and initiates the mechanisms of anaerobic metabolism. As a result, the process of mitochondrial apoptosis is triggered, and the cytoskeleton is degraded by proteolytic enzymes [2,3,4].

Restoration of vascular circulation results in the generation of reactive oxygen species (ROSs) [5] and reactive nitrogen species (RNSs) [6]. The ROSs and RNSs react with proteins, carbohydrates, lipids, and nucleic acids [7]. The resultant chemical modifications disrupt cellular homeostasis and impair the physiological processes, thus becoming the main cause of reperfusion injury [8]. The above processes are accompanied by the development of inflammation, intensified by the activation of leukocytes [9].

The described pathological phenomena are responsible for the development of an ischemia reperfusion injury (IRI). IRI has detrimental effects on vessels and microcirculation. In small arteries, IRIs lead to disturbances in vasodilation, which is mediated by nitric oxide (NO). In the terminal arterioles, IRIs are characterized by the fluid passing beyond the basal membrane of endothelial cells and by the reduction in the number of capillaries with active blood flow [10].

The authors of the following paper wanted to assess how IRI affects arterial vasoconstriction, using an experimental model assessing the reactivity of the vessel under the influence of phenylephrine (Phe), a compound that is a synthetic sympathomimetic. An important element of this designed study was the assessment of how contractility is modified by cyclic nucleotides (CNs) and calmodulin in ischemia and reperfusion.

CNs, cyclic adenosine monophosphate (cAMP), and cyclic guanosine monophosphate (cGMP) are involved in the regulation of vascular function and act as secondary transmitters, e.g., for NO [11]. The effects of cAMP and cGMP depend on the activity of adenylate cyclase (AC) and guanylate cyclase (CG), the quantitative ratio between the two compounds, and the levels of their degradation by cyclic phosphodiesterases (cPDEs) [12]. The increase in the intracellular levels of cAMP and cGMP leads to vasodilation in a complex mechanism based on phosphorylation of proteins which control the vessel wall tone, with the involvement of protein kinases of types A (PKA) and G (PKG), as well as AKAP and EPAC proteins. In addition, the increase in the cAMP and cGMP levels inhibits the proliferation and migration of vascular smooth muscle cells (VSMCs) [13].

Calmodulin is a ubiquitous and multifunctional [14] calcium ion-binding protein that mediates the regulatory effects of Ca^2+^ ions. In smooth muscle cells, calmodulin is responsible for initiating interactions between actin and myosin filaments in response to increased concentrations of Ca^2+^, by means of activating the smooth muscle myosin light chain kinase (smMLCK) and phosphorylating the myosin regulatory light chain (MRLC) [15]. The process is modulated by a separate calmodulin-dependent kinase known as Ca^2+^/calmodulin-dependent protein kinase II (CaMKII). CaMKII phosphorylates smMLCK, resulting in an increase in calmodulin levels, which is required for activation of myosin light chain kinase, and may thus be responsible for desensitization of the contractile response to Ca^2+^ ions [16]. Another potential level for calmodulin’s regulation of smooth muscle contractions is related to proteins that control the intracellular movement of Ca^2+^ ions, i.e., inositol triphosphate receptors (IP3Rs) and ryanodine receptors (RyRs), as well as proteins which indirectly regulate Ca^2+^ ions by synthesizing and degrading CNs, such as phosphodiesterases (PDEs) [17].

## 2. Results

### 2.1. Arterial Incubation in the Presence of Forskolin

A decrease in arterial reactivity was observed in the presence of forskolin under control conditions. The recorded values of P_0_ and P_end_ were 1.58 mmHg and 66.67 mmHg, respectively. During reperfusion, the reactivity is significantly increased, with P_1_ reaching 107.86 mmHg. The value of E_rel_ increases significantly, along with the EC_50_ value decreasing significantly to 3.96 × 10^−8^ M/L (Table 1). During reperfusion, the shape of the curve is similar to the control curve for Phe. The curve is shifted towards higher values E_a_/E_max_ × 100% (Figure 1).

### 2.2. Arterial Incubation in the Presence of A7

A significant decrease in arterial reactivity and a reduction in E_max_ were observed in the presence of A7 in the control phase. In the reperfusion phase, the increase in reactivity was observed as expressed by increasing E_max_. With regard to the control conditions, no significant differences were observed in the reactivity of VSMCs at lower Phe concentrations. The differences became evident only at concentrations higher than 10^−7^ M/L. In the reperfusion phase, a significant increase in P_end_ was observed to the value of 89.26 mmHg as opposed to control conditions in which no significant differences were observed between mean P_end_ values (Table 1). Restoration of baseline vascular reactivity was observed during reperfusion. However, reperfusion is characterized by a flattening and shift of the sigmoidal curve to the right towards higher Phe concentrations (Figure 2).

## 3. Discussion

With regard to the measurements of arterial reactivity upon incubation with forskolin, the control phase was characterized by a flattening of the concentration effect curve (Figure 1) and a reduction in arterial reactivity, along with the reduction in E_rel_. VSMC contraction is observed only at higher Phe concentrations (>10^−8^ M/L). The P_end_ and E_rel_ values increase significantly during the reperfusion phase (Table 1). The observed effect is somewhat surprising as the increased VSMC contractility is already observed at very low Phe concentrations (>10^−10^ M/L) and in the presence of an AC activator within the extravascular space. The increase in AC activity results in vasodilation in the control phase as opposed to vasoconstriction in the reperfusion phase. In order to elucidate the observed phenomena, they should be considered in the context of the entire pathophysiological mechanism of IRI, which is characterized by a strong inflammatory response and ischemic tissue being exposed to deleterious factors. Inflammation within the perivascular area is manifested, e.g., by the increased macromolecular permeability within microcirculation vasculature, particularly postcapillary venules [18]. Restoration of perfusate circulation during the reperfusion phase creates conditions for AC stimulation, hence the increase in cAMP levels. Although surprising, the vasoconstriction effect is linked to the activation of compensatory mechanisms within the highly active VSMCs. During reperfusion, increasing cAMP levels activate the EPAC protein, which modulates the activity of the Rap proteins. EPAC proteins control numerous biological [19] and cellular processes, including migration, proliferation, and apoptosis [20,21]. They stimulate the formation of intercellular connections within the endothelium to restore vascular integrity and reduce vascular permeability [22]. EPAC1 was shown to protect the retina against the consequences of ischemia in the mechanism of exerting its effect on endothelial mediators of inflammation [23]. Other tests in animal models showed that upon micro-pharmacotherapy aimed at inhibition of EPAC1, the resulting inhibition of Rap1 exerts a protective effect against the negative outcomes of IRIs due to the suppression of apoptosis [24] or to a multifactorial protective impact on IRI-affected cells [25]. Although the aforementioned studies focused on the potential prevention of the consequences of an IRI, they were carried out in cardiomyocytes and did not directly evaluate the contractile properties of VSMCs. Based on the available data, the activity of EPAC proteins in ischemia should be considered to be related to improved vascular contractility in the mechanism of improving vascular wall integrity.

In the analysis of the reactivity of arteries incubated with A7 hydrochloride following the experimental design, a significant increase in P_end_ and E_rel_ was observed in the reperfusion phase (Table 1), despite the initially expected decrease in arterial reactivity during the control phase, which corresponded to the inhibition of calmodulin activity. The reperfusion phase is characterized by the promotion of calmodulin-independent vasoconstriction mechanisms, resulting from satisfactory homeostasis of VSMCs being achieved in the established experimental conditions. These mechanisms have no impact on the EC_50_ value and are observed at higher Phe concentrations (>10^−7^ M/L) (Table 1). Low activity of contraction restricting Ca^2+^/calmodulin-dependent mechanisms is observed.

The active Ca^2+^/calmodulin complexes exert pleiotropic effects and affect the receptor- and non-receptor-based mechanisms, including the activity of Ca^2+^/calmodulin-dependent protein kinase I and II—the good condition of VSMCs results from the inactivity of CaMKII, which modulates their contractile properties. The activity of CaMKII during reperfusion enhances calcium overload as the key factor responsible for ischemia-reperfusion injuries, and thus aggravates the disruption of VSMC homeostasis, including the cellular vasoconstriction potential. This pathomechanism results from the activation of Na^+^/H^+^ and Na^+^/Ca^2+^ ion exchangers. As shown by one of the studies on the scope of CaMKII activity within the myocardium, CaMKII phosphorylates effector proteins to regulate ionic homeostasis, contraction, inflammatory response, and apoptosis of cardiomyocytes during reperfusion [26], and pharmacological inhibition of CaMKII may improve the function of cardiomyocytes and reduce the level of cell apoptosis during reperfusion by affecting the levels and function of mitochondrial calcium [27]. The overall effect of CaMKII may vary and depend on the calmodulin inhibition rate, as well as on the activity of calcium/calmodulin complexes and levels of calmodulin-dependent protein kinases synthesized in the positive feedback mechanism.

## 4. Materials and Methods

This study was conducted on a group of 30 male Wistar rats (mean age of 3 months, body weight of 250–350 g). Pharmacometric analyses were performed on isolated tail arteries, along with vascular endothelium as a resistive artery model. The animals were kept in a 12 h day/night cycle. The temperature within the animal housing facility was maintained at 20–21 °C, and the humidity was in the range of 50–60%. The animals were supplied with ad libitum food and water during the day and night. Rats were euthanized with urethane anesthesia.

### 4.1. Artery Preparation

Following the removal of tissues surrounding the artery, the proximal non-bifurcating segment was cannulated over a length of approximately 2–3 cm and placed vertically inside a test vessel, having an approximate volume of 20 mL. The vessel was experimentally loaded with a pre-weighted mass of about 0.5 g, and stabilized in aerated Krebs fluid at 37 °C and pH 7.4 for 2 h before the start of the test. The artery was connected via the cannula to a perfusion system measuring and recording the perfusion pressure. Perfusate circulation was achieved by means of two peristaltic pumps, their throughput gradually increasing from 0.25 to 1 mL/min until a pressure of between 2 and 4 kPa had been reached. The outcome measure consisted of the difference between the final pressure (P_end_) and the initial pressure (P_0_) within the designed system.

### 4.2. Study Design

This study was divided into phases and performed in stages. Each phase consisted of a preparatory part (part one) and an executive part (part two). In the first part, a segment of the artery was prepared according to the procedure described in Section 4.1. Before starting the second part, compounds modifying contractility were added to the test vessel. In the second part, a stable perfusion pressure between 2 and 4 kPa was obtained using a peristaltic pump, then Phe was introduced into the test vessel in increasing concentrations, and the pressure was recorded. Individual measurements were taken in series, with each series consisting of the assessments of perfusate flow following increasing doses of Phe being administered to the extravascular space in the presence of modifiers, after the pressure had stabilized following the administration of the previous lower dose. A similar method had been previously used with success, e.g., for the assessment of rho-kinase inhibition [28]. It should be noted that before performing the series of tests in the second part, the control curve was determined for Phe without the presence of modifying substances in the extravascular space. This procedure was aimed at determining the standard curve and was validation for the created experimental model. The presence of the standard curve is included in the graphs in the section describing the results.

### 4.3. Compounds Used in the Extravascular Space

Forskolin (an AC activator) and A7 hydrochloride (a strong antagonist of calmodulin active in the calmodulin-activated PDE activity inhibition mechanism) were used for administration into the extravascular space.

### 4.4. Research Phases

This study was divided into 3 phases. Phase 1 (the control phase) involved the assessment of arterial contractility as triggered by Phe in the presence of modifiers within the extravascular space. Phase 2 (the ischemia phase) consisted of measurements being carried out as in Phase 1, albeit after 60 min of perfusate flow suppression. In Phase 3 (the reperfusion phase), the flow parameters were evaluated again 30 min after the restoration of perfusate flow. These three phases of the experiment were preceded by a calibration curve being determined on the basis of flow parameters measured for Phe alone as present within the extravascular space.

### 4.5. Statistical Analysis

Statistical analyses were carried out using the Statistica 13.1 software. The Shapiro–Wilk test was used to evaluate the compliance of data distribution with the normal distribution pattern. Differences between groups were compared by means of ANOVA. The statistical significance level was established at *p* < 0.05. The concentration effect curves were determined with the use of van Rossum’s cumulated concentration method [29], as amended by other researchers. The agonist concentration responsible for achieving 50% of the maximum effect (E_max_) was defined as EC_50_. The relative effect parameter (E_rel_), expressed as the ratio of the observed effect to the reference effect consisting of the contractility curve parameters measured in response to the vasopressor activity of Phe, was also used in the assessment of the experiment results.

## 5. Conclusions

The study methodology was developed in accordance with the classical pharmacometric methods, with concentration effect curves determined according to the van Rossum cumulative concentration method. It was shown that the physiological effects of AC and calmodulin observed in the control phases of both parts of the experiment might not be identical to those of these compounds during reperfusion and that, contrary to physiological conditions, cAMP in reperfusion assumes vasoconstrictor characteristics. The obtained results were analyzed in the context of the available literature in this field, indicating that the modification of AC and calmodulin activity results from changes in VSMC homeostasis, and the action of these compounds should be considered in light of the entire pathophysiology of IRI. This study did not consider the effect of membrane calcium channels on vascular endothelial function. It is reasonable to conduct further studies evaluating the function of phospholipase C, V1 receptor, and other metaboreceptors in IRI. Analyzing the importance of cellular signaling pathways involving various second messengers in reperfusion may contribute to introducing changes in the clinical practice of treating diseases in which the pathophysiological basis is IRI.

## Figures and Tables

**Figure 1 ijms-24-10355-f001:**
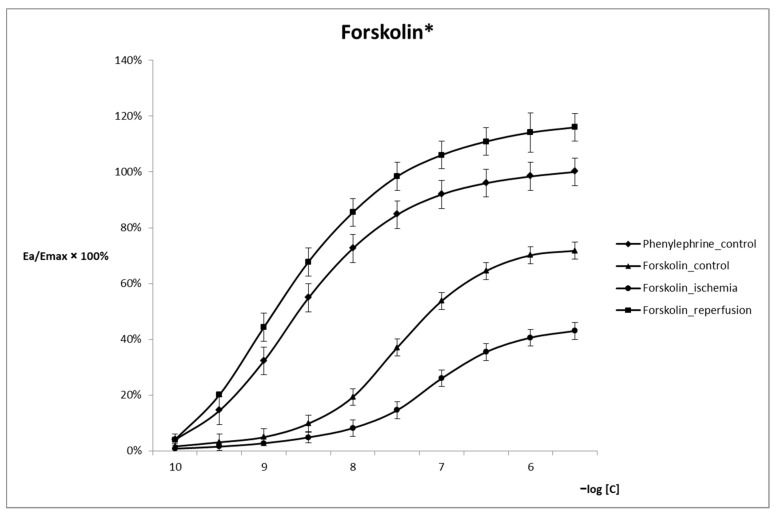
Reactivity of arterial smooth muscle tissue following incubation with forskolin as measured under control, ischemia, and reperfusion conditions. * The curves differ significantly from the control within the E_max_ range of 20–80% (*p* < 0.05).

**Figure 2 ijms-24-10355-f002:**
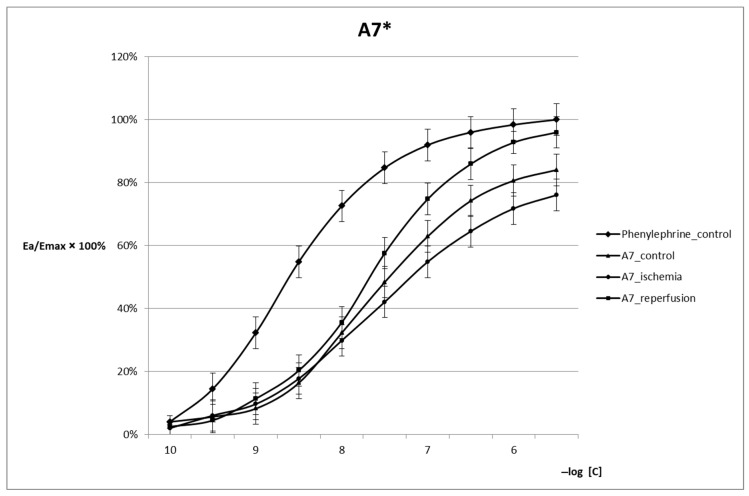
Reactivity of arterial smooth muscle tissue following incubation with A7 as measured under control, ischemia, and reperfusion conditions. * The curves differ significantly from the control within the E_max_ range of 20–80% (*p* < 0.05).

**Table 1 ijms-24-10355-t001:** Values of n, P_0_, P_end_, E_rel_, EC_50,_ and R_p_ for phenylephrine in the presence of forskolin and A7 at the concentration of 2 × 10^−4^ M/L under control conditions after 60 min of ischemia and during reperfusion.

Parameter	Test Conditions					
	Control		60 min Ischemia		Reperfusion	
	Forskolin	A7	Forskolin	A7	Forskolin	A7
*n* ^1^	24	22	29	23	25	27
*P_0_* [mmHg] ^2^	1.58	3.16	0.83	2.80	3.83	2.33
*P_end_* [mmHg] ^3^	66.76	78.16	40.05 *	70.73	107.86 **^/a^	89.26 **
*E_rel_* [%] ^4^	72	84	43 *	76 *	116 **^/a^	96 **
*EC_50_* [M/L] ^5^	$4.14\times 10^{-7}$	$6.26\times 10^{-8}$	$6.79\times 10^{-7}$	$3.74\times 10^{-7}\;$*	$3.96\times 10^{-8\;}$**^/a^	$6.82\times 10^{-8}$ ^a^
*R_p_ * ^6^	0.11	0.77	0.73	1.33	1.20	0.70

^1^ The number of concentration-versus-effect curves used for calculation; ^2^ initial pressure expressed in mmHg; ^3^ final pressure expressed in mmHg; ^4^ relative effect; ^5^ compound concentration required to achieve 50% of the maximum effect; ^6^ relative potency expressed as the quotient of EC_50_ for phenylephrine under control conditions and the EC_50_ of interest; differences were considered statistically significant for *p* < 0.05 (* control vs. 60′ ischemia; ^a^ 60′ ischemia vs. reperfusion; ** control vs. reperfusion).

## Data Availability

The data presented in this study are available on request from the corresponding author. The data are not publicly available as they are the property of the Department of Pharmacology and Therapeutics, Faculty of Medicine, Collegium Medicum in Bydgoszcz, Nicolaus Copernicus University in Toruń, 85-094 Bydgoszcz, Poland.

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
