# Peer review of "Cyclic AMP but Not Calmodulin as a Potential Wasoconstrictor in Simulated Reperfusion"

_ijms, 2023, doi:10.3390/ijms241210355_

Round 1

Reviewer 1 Report

The article is presented in quite unusual form. After a very long introduction, which presents well-known facts, the presentation of the results begins (section 2). However, without a description of the method of experiments, this section is not understandable, and the presented graphic dependencies (Figures 1 & 2) look completely meaningless, since it is unclear what served as a constrictor. Also it is not clear what is “Reactivity of arterial smooth muscle tissue”. Only after reading section 4, which follows the discussion, the results may be understood. Reading such an article is extremely inconvenient.

The methods are not described in detail. In particular, it is not clear how the cut segment of the artery was given its normal length and what was the tension of the cut segment of the vessel. These data are important because they largely determine the mechanoceptive function of the endothelium. The perfusion scheme and the procedure of pressures recordings is also not described in detail. I think than section “Materials and Methods” should be rewritten, so that the reader can clearly understand not only the general scheme of experiments but the particular details, as well.

The abstract of the article also contains a lot of introductory information, but does not contain clear conclusions of the research.

In my opinion, the presented text of the article needs a radical revision.

Reviewer 2 Report

The present study is full of flaws and not well written or executed hence cannot be recommended for publication in this journal. Some of the major points are as follows:

The introduction section is not very well written.

There is no explanation for the rationale of the study and what problem the authors were trying to address.

There is no aim defined in the introduction and the results were abruptly introduced and written.

Results are poorly written and no reference to figures or table they are referring to.

There is no clear evidence that calmodulin has no role in simulated reperfusion and only cAMP plays the major role, hence need more experiments to establish this.

Conclusion needs to be rewritten to explain the findings and the benefits of the study.

There are no limitations defined for the study.

English language/grammar needs to be checked thoroughly.

There are few grammatical errors in the documents and hence the authors are advised to read the document carefully.

Reviewer 3 Report

The study was conducted on a group of 30 male Wistar rats. This number of sample is considered as small. Would consider this as a pilot study.

Round 2

Reviewer 1 Report

I think that the manuscript has been improved and may now be published in the IJMS (although I still have some questions concerning the preservation of vascular endothelium during the prteparation procedure.) 

Reviewer 2 Report

The authors have incorporated all the suggestions from the reviewer in their manuscript and edited the text accordingly and hence can be published now.